# Optimized Radial Basis Function Neural Network Based Intelligent Control Algorithm of Unmanned Surface Vehicles

**Renqiang Wang** [1,*], **Donglou Li** [2] **and Keyin Miao** [1]

1   Navigation College, Jiangsu Maritime Institute, Nanjing 211170, China; miaoky19692003@163.com
2   Maritime College, Hainan Vocational College of Science and Technology, Haikou 571126, China; captli15902759889@163.com
*   Correspondence: wangrenqiang2009@126.com; Tel.: +86-180-688-14286

**Abstract:** To improve the tracking stability control of unmanned surface vehicles (USVs), an intelligent control algorithm was proposed on the basis of an optimized radial basis function (RBF) neural network. The design process was as follows. First, the adaptation value and mutation probability were modified to improve the traditional optimization algorithm. Then, the improved genetic algorithms (GA) were used to optimize the network parameters online to improve their approximation performance. Additionally, the RBF neural network was used to approximate the function uncertainties of the USV motion system to eliminate the chattering caused by the uninterrupted switching of the sliding surface. Finally, an intelligent control law was introduced based on the sliding mode control with the Lyapunov stability theory. The simulation tests showed that the intelligent control algorithm can effectively guarantee the control accuracy of USVs. In addition, a comparative study with the sliding mode control algorithm based on an RBF network and fuzzy neural network showed that, under the same conditions, the stabilization time of the intelligent control system was 33.33% faster, the average overshoot was reduced by 20%, the control input was smoother, and less chattering occurred compared to the previous two attempts.

**Keywords:** intelligent computing; RBF neural network; genetic algorithms; sliding mode control; USV; optimized

## 1. Introduction

Since the concept of unmanned autonomous ships (UASs) was established, the research and development of UASs has become a key breakthrough project in the global shipping industry. To this end, the research and development institutions in various countries have conducted in-depth research on the intelligent navigation and control of UAS. At present, research on intelligent motion control navigation is in full swing. The most representative applications are various types of small unmanned surface vehicles (USVs) [1]. Developing a small intelligent hull and conducting a series of studies will lay a solid foundation for future large-scale intelligentization.

It is well known that steering tracking control is an important research hotspot in the field of ship motion and control, which is related to both the economics and safety of ship navigation [2,3], such as in ship collision avoidance [4,5]. Due to the uncertain parameters, nonlinearity, and external interference [6] in the USV motion control model, conventional proportional–integral–derivative (PID) control is not effective enough, and the control input chattering situation is serious [7]. Therefore, ideal sliding mode control (ISMC) [8,9] can be effectively applied to the motion control of nonlinear systems. Neural network systems have universal approximation performance for arbitrary functions, which

can effectively solve the problem of the uncertainty of USV motion systems [10,11]. Wang et al. [12] proposed the use of a radial basis function (RBF) network for control input compensation and designed an intelligent tracking control algorithm for USVs based on SMC under limited input conditions. Dongdong et al. [13] proposed an adaptive trajectory tracking control strategy for underactuated unmanned surface vehicles. The neural network minimum learning parameter method proposed in this paper features a small amount of computation. After combining the SMC and RBF networks and applying them to the design of USV motion control, the performance of the controller could be further improved [14,15].

However, when neural networks are used for the modeling and control of motion systems [16], the function estimation and classification functions of neural networks are most commonly used. The key to designing a neural network is to determine the structure and connection weight coefficients of the neural network. For the most typical radial basis function (RBF) neural network, the connection weights and parameters of the Gaussian functions are mainly determined by experience [17]. If the parameters are not properly selected, it is easy for the system to fall into local extremes [18]. Genetic algorithms (GAs) can be used for calculation optimization [19] and in the design of neural networks [20]. In [21], a GA optimized the center vector and basis width of an RBF neural network and updated the weight of the RBF network in real time, improving the accuracy of RBF tracking. In [22], a genetic algorithm was used to optimize the weight of a RBF network and a clustering algorithm was used to determine the number of centers of the basis function of the RBF network. Then, a coke quality prediction model based on the GA-optimized RBF neural network was designed. Based on the literature [17], the key to RBF network application is to determine its three parameters.

Considering the shortcomings of GAs [17], some improvements have been proposed, such as an adaptive mutation algorithm and an excellent individual protection algorithm. In [23], utilizing the excellent global search ability of gene expression programming (GEP) to optimize the RBF neural network, a GEP-optimized RBF multi-output model algorithm (GEP-RBF algorithm) was designed. This model has good prediction accuracy and multi-output balance.

Motivated by the above-mentioned observations, an intelligent control algorithm was proposed based on SMC with an optimized RBF network. GA was used to optimize the RBF network, and the optimized RBF network was directly approximated to the USV motion control input instructions. Then, the system stabilization function was constructed using SMC theory. Finally, the USV navigation motion intelligent control algorithm was introduced using stability theory.

The contributions of this paper can be summarized as follows. The previous USV motion controller is remodeled based on SMC and RBF Neural Network (RBFNN). The GA-optimized RBF network directly inputs the control law, which overcomes the chatter caused by the uncertainty term in the USV model. At the same time, by modifying the adaptation value and mutation probability and by differentiating N subpopulations, the improvement of the general GA algorithm can reduce the prevalence of falling into local extremes, speed up the optimization process, and further improve the input accuracy of the RBF network.

## 2. RBF Neural Network and Its Genetic Optimization

The RBF neural network is a neural network proposed by Moody and Darken [17] in the late 1980s. It is a three-layer, feed-forward network with a single hidden layer, as shown in Figure 1. Because it simulates a neural network structure that is locally adjusted in the human brain and covers the receiving area with itself, the RBF network is a local approximation network. The fact that the mapping from input to output is nonlinear and the mapping from the hidden layer space to the output space is linear can greatly speed up the learning process and avoid local minima and can also approximate any continuous function with arbitrary precision [17]. However, it is difficult to determine parameters like connection weights, center widths, and center values for the Gaussian function [18].

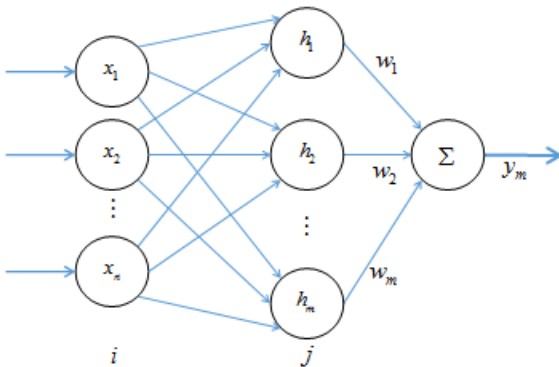

**Figure 1.** The structure of the radial basis function (RBF) neural network.

### 2.1. RBF Neural Network Approximation Algorithm

The RBF network input/output algorithm [17] is

$$
\begin{cases}
h_j = \exp\left(\dfrac{\|x - c_j\|^2}{2b_j^2}\right) \\
\delta = W^{*\mathrm{T}} h(x) + \varepsilon
\end{cases}
\tag{1}
$$

where $x$ is the network input, $i$ is the i-th input of the network input layer, $j$ is the j-th network input of the network hidden layer, $h = \left[h_j\right]^{\mathrm{T}}$ is the output of the Gaussian function, $W^*$ is the ideal weight of the network, and $\varepsilon$ is the error of the ideal neural network approximating $\delta$, $\varepsilon \le \varepsilon_{\max}$.

If the network input is taken as $x = [x_1\ x_2 \cdots x_n]^{\mathrm{T}}$, then the output of the network is

$$
\hat{f}(x) = \hat{W}^{\mathrm{T}} h(x)
\tag{2}
$$

where $\hat{f}$ is the network output and $\hat{W}$ is the estimated weight of the neural network.

### 2.2. Optimization of the RBF Neural Network based on the Genetic Algorithm

The current methods will be used to improve the performance of RBF networks, which include fuzzy algorithms and intelligent optimization algorithms [24–27]. In fuzzy systems, the design of fuzzy sets, membership functions, and fuzzy rules is based on empirical knowledge, and the algorithm itself does not have the ability to learn autonomously [25]. However, the intelligent optimization algorithm represented by the genetic algorithm [26,27] can realize self-learning by using the law of biological evolution, ultimately converging to the most adaptive group to obtain the optimal solution or most satisfactory solution.

In addition, the main advantages of genetic algorithms [17] are listed below:

(1) To solve any form of objective function and constraint optimization problem, whether it is linear or nonlinear, discrete or continuous, the genetic algorithm does not require any mathematical model. Depending on its evolutionary nature, the inherent nature of the problem need not be known during the search process.

(2) The ergodicity of evolutionary operators makes genetic algorithms very efficient in globally searching for probabilistic meanings.

(3) For a variety of special problems, genetic algorithms can provide great flexibility to mix and construct domain-independent heuristics, thereby ensuring the effectiveness of the algorithm.

When the GA runs its optimization calculations, a large system size will cause poor optimization performance. Similarly, a system sample without important characteristic genes can also cause the optimization process to converge prematurely and fail to reach the optimal solution [28]. In order

to solve the above problems, this paper adopts a modification to the adaptation value and mutation probability to improve the traditional optimization algorithm.

Thus, the improved genetic algorithm (IGA) is used to determine the connection weight parameters of the RBF network. The genetic optimization process of the RBF neural network is shown in Figure 2, and the process of optimization can be described as follows:

**Step 1:** The N subpopulations are initialized, and the initial network parameters (connection weights, center width, and center value of the Gaussian function) are encoded as genes.

**Step 2:** N subpopulations carry out evolutionary operations independently.

**Step 3:** Performance judgement? If no, go to the next step. If yes, end, and go to the RBF learning step.

(1) Firstly, autonomous learning of the RBF network.
(2) Secondly, absolute error calculation.
(3) Thirdly, parameter updating.
(4) Finally, performance judgement? If Yes, end. If No, go to autonomous learning and error calculation.

**Step 4:** The average fitness of N subpopulations is calculated.

**Step 5:** The selection and cross operations are performed separately.

**Step 6:** Mutation operations are performed separately.

**Step 7:** The new N subpopulations are recalculated and returned to Step 2.

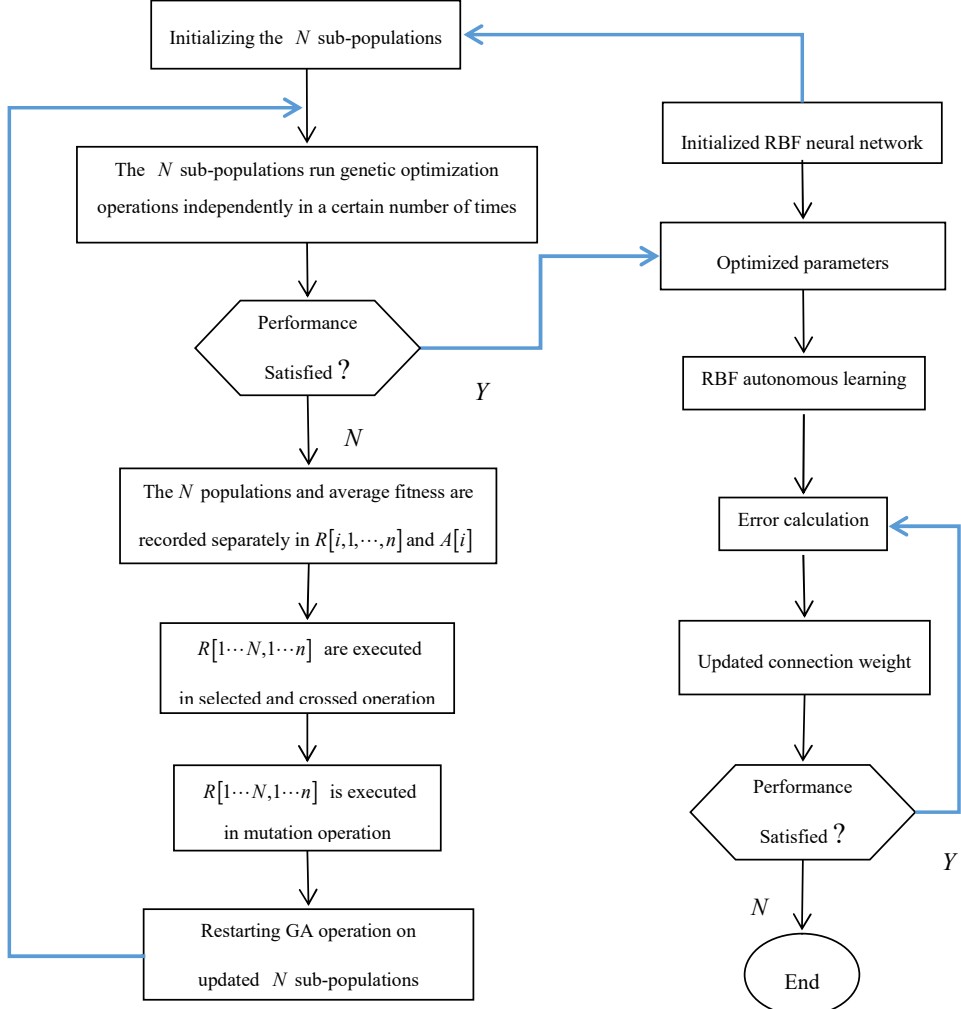

**Figure 2.** The process of the neural network optimization with the improved genetic algorithm.

### 2.2.1. Adaptation Value Correction

If $f$ is the adaptation value calculated in the usual way, and $\overline{f}$ is the average adaptation value [17], then the revised adaptation value is

$$f' = \begin{cases} k_1 \cdot \overline{f} & f \geq k_1 \cdot \overline{f} \\ f & f < k_1 \cdot \overline{f} \end{cases} . \tag{3}$$

Here, $k_1 > 1$. The effect of this correction is to reduce the influence of genes whose adaptation value is too large, slow down their convergence speed, and expand the search space.

According to the model theory, after the adaptation value is modified, as the genetic algorithm gradually evolves, the average adaptation value will become increasingly larger, and optimization will develop in an ideal direction.

### 2.2.2. Correction of Mutation Probability

The mutation probability is corrected as follows:

$$p_m(i+1) = \begin{cases} p_{mh} & \text{The highest adaptation value} \\ & \text{is canstant in N generations} \\ p_m(i) & k_2 p_m(i) \leq p_{ml} \\ k_2 p_m(i) & others \end{cases} \tag{4}$$

where $i$ represents the algebra of the GA's evolution; $p_{mh}$ and $p_{ml}$ represent the upper and lower limits of the mutation probability; and $k_2$ is a constant less than 1.

The correction method can be used to increase the mutation probability automatically when premature convergence occurs to expand the search space. At the same time, in order to prevent the best results being destroyed due to an overly high mutation probability, the best samples are kept in each iteration. Generally, the values of the above parameters are: $p_{nh} \in [0.5, 1]$, $p_{ml} \in [0, 0.1]$, $k_1 \in [1, 10]$, $k_2 \in [0.8, 1]$, and $G \in [1, 1000]$.

In order to obtain satisfactory approximation accuracy, an absolute error index is used as the minimum objective function for parameter selection:

$$J = 100 \sum_{i=1}^{N} |e(i)| \tag{5}$$

where $N$ is the total approximation step, and $e(i)$ is the approximation error of the i-th step.

### 3. USV Ideal Sliding Mode Control Based on the RBF Neural Network

Because the SMC control law contains the uncertainty terms $f(x_2)$ and $g$, the optimized RBF neural network is used to approximate the control input instruction to achieve ideal control of the USV. First, the initial network parameters (connection weights, center width, and center value of the Gaussian function) are encoded as genes, and the optimized parameters output by the IGA are accepted. Then, the absolute error of the control system is determined by the optimized RBF network. Then, the RBF network output is transmitted to the SMC control unit after performing autonomous learning. Lastly, an intelligent control algorithm for USV navigation motion is introduced based on the SMC method with Lyapunov stability theory. The intelligent control algorithm flow is shown in Figure 3.

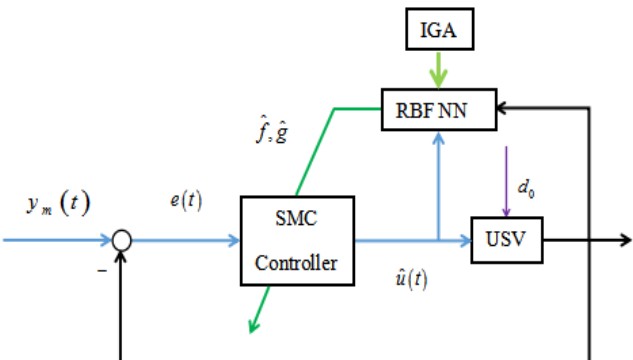

**Figure 3.** The intelligent control algorithm flow of the unmanned surface vehicle (USV).

### 3.1. USV Motion Mathematical Model

In the case of external disturbances and system parameter disturbances, Nomoto's equation is used as a dynamic mathematical model of USV plane motion [29,30]:

$$\ddot{\varphi} + \frac{H(\dot{\varphi})}{T + \Delta T} = \frac{K + \Delta K}{T + \Delta T}\delta + d(t) \tag{6}$$

where $K$ and $T$ are the ship's parameters, which can be obtained through ship experiments, and $\Delta K$ and $\Delta T$ are the perturbations. $\varphi$ is the heading and $\delta$ is the rudder angle. $d(t)$ is the external disturbance, $d(t) \leq d_{\max}$; its equivalent rudder angle interference model in Nomoto's model will be explained in the computer experiments section. $H(\dot{\varphi})$ is a nonlinear function of $\dot{\varphi}$, which can be approximated as

$$H(\dot{\varphi}) = \alpha_1\dot{\varphi} + \alpha_2\dot{\varphi}^3 + \alpha_3\dot{\varphi}^5 + \cdots \tag{7}$$

where $\alpha_i (i = 1, 2, 3, \cdots)$ is a real-valued constant.

In order to perform sliding mode control design, Equation (1) needs to be transformed into the following form:

$$\begin{cases} \dot{x}_1 = x_2 \\ \dot{x}_2 = f(x_2) + g \cdot u + d(t) \\ y = x_1 \end{cases} \tag{8}$$

where $x_1 = \varphi$, $x_2 = \dot{\varphi}$, $u = \delta$, $f(x_2) = -\frac{1}{T + \Delta_T}H(\dot{\varphi})$, and $g = \frac{K + \Delta_K}{T + \Delta_T}$.

### 3.2. Ideal Sliding Mode Control Based on the RBF Neural Network

The USV navigation controller is introduced with SMC technology and Lyapunov stability theory [31]. The optimized RBF network is applied to approximate the control law of USV with the problem of system parameter perturbation.

According to the SMC design theory, the sliding mode surface function is defined as follows:

$$s = \lambda e + \dot{e} \tag{9}$$

where $\lambda > 0$, $e = x - x_d$, $x$ is the actual system output, and $x_d$ is the system input instruction.

Thus, Equation (10) can be inferred using Equation (9):

$$\dot{s} = \lambda\dot{e} + \ddot{e} = f(x_2) + [v + d(t)] + g \cdot u \tag{10}$$

where $v = -\ddot{x}_d + \lambda\dot{e}$.

The existence of an ideal sliding mode control law [32] in the absence of external interference and system parameter perturbation can make the controlled system (Equation (8)) globally stable, and the system convergence speed can be adjusted by parameter $\varepsilon$. The ideal sliding mode control law is

$$u^* = -\frac{1}{g}[f(x_2) + v] - \left(\frac{1}{\varepsilon g} + \frac{1}{\varepsilon g^2} - \frac{\dot{g}}{2g^2}\right)s. \tag{11}$$

The control law has limitations because the system function $f(x_2)$ and gain $g$ are unknown. The optimized RBF neural network was used to approximate $u^*$ to achieve ideal control of USV navigation. The inputs of the RBF neural network, containing the system variables and sliding mode surface functions, are

$$z = \left[x^T \quad s \quad \frac{s}{\varepsilon} \quad v\right]^T \in \Omega_z \subset R^5 \tag{12}$$

where the compact set is expressed as

$$\Omega_z = \left\{\left(x^T \quad s \quad \frac{s}{\varepsilon} \quad v\right) \middle| x \in \Omega; \; x_d \in \Omega_d\right\}. \tag{13}$$

The ideal output of the RBF neural network can be rewritten [18] as

$$u^*(z) = W^{*T}h(z) + \pi_l, \; \forall z \in \Omega_z \tag{14}$$

where $W^* = \arg \min_{W \in R^l} \left\{\sup_{z \in \Omega_z}\left|W^T h(z) - u^*(z)\right|\right\}$, $h(z)$ is a Gaussian function, $\pi_l$ is a network approximation error, and $|\pi| \leq \pi_0$.

After being approached by the GA-optimized RBF neural network, the actual input of the SMC controller was

$$u = \hat{W}^T h(z) \tag{15}$$

In this formula, $\hat{W}$ is an estimated value of $W^*$.

The network learning adaptive law is designed as follows:

$$\dot{\hat{W}} = -\Gamma\left(h(z)s + \sigma\hat{W}\right) \tag{16}$$

where $\Gamma = \Gamma^T$ and $\sigma > 0$.

Equation (17) can then be obtained according to Equations (10) and (15):

$$\dot{s} = f(x_2) + [v + d(t)] + g \cdot \hat{W}^T h(z). \tag{17}$$

Similarly, Equation (18) is produced according to Equations (14) and (17):

$$\dot{s} = f(x_2) + [v + d(t)] + g \cdot u^*(z) + g \cdot \left(\hat{W}^T h(z) - W^{*T} h(z) - \pi_l\right). \tag{18}$$

In order to suppress the external disturbance of the system, the ideal sliding mode control law in Equation (11) is further improved as

$$u^{*'} = -\frac{1}{g}[f(x_2) + v + \eta\text{sgn}(s)] - \left(\frac{1}{\varepsilon g} + \frac{1}{\varepsilon g^2} - \frac{\dot{g}}{2g^2}\right)s \tag{19}$$

where $\eta \geq d_{\max}$.

Further, Equation (20) can be deduced by Equations (18) and (19):

$$\dot{s} = g \cdot \left(\tilde{W}^T h(z) - \pi_l\right) + (d(t) - \eta\text{sgn}(s)) - \left(\frac{1}{\varepsilon g} + \frac{1}{\varepsilon g^2} - \frac{\dot{g}}{2g^2}\right)s \tag{20}$$

where $\tilde{W} = \hat{W} - W^*$.

Based on this, the Lyapunov Function was constructed as follows:

$$V = \frac{1}{2}\left(\frac{s^2}{g} + \tilde{W}^T \Gamma^{-1} \tilde{W}\right).$$

(21)

Thus, Equation (22) can be inferred via Equation (21):

$$
\begin{aligned}
\dot{V} &= \frac{s\dot{s}}{g} - \frac{\dot{g}}{2g^2}s^2 + \tilde{W}^T \Gamma^{-1} \dot{\hat{W}} \\
&= \frac{s}{g}\left[g\left(\tilde{W}^T h(z) - \pi_l\right) - \left(\frac{1}{\varepsilon} + \frac{1}{\varepsilon g} - \frac{\dot{g}}{2g}\right)s\right] - \frac{\dot{g}}{2g^2}s^2 + \tilde{W}^T \Gamma^{-1}(-\Gamma)\left(h(z) + \sigma\hat{W}\right) \\
&= -\left(\frac{1}{\varepsilon g} + \frac{1}{\varepsilon g^2}\right)s^2 - \pi_l s - \sigma \tilde{W}^T \hat{W} + \frac{1}{g}(d(t)s - \eta[s]) \\
&\leq -\left(\frac{1}{\varepsilon g} + \frac{1}{\varepsilon g^2}\right)s^2 - \pi_l s - \sigma \tilde{W}^T \hat{W}
\end{aligned}
$$

(22)

The inequality equations are as follows:

$$
\begin{cases}
2\tilde{W}^T \hat{W} &= \tilde{W}^T(\tilde{W} + W^*) + (\hat{W} - W^*)^T \hat{W} \\
&= \tilde{W}^T \tilde{W} + (\hat{W} - W^*)^T W^* + \hat{W}^T \hat{W} - W^{*T} \hat{W} \\
&= \left\|\tilde{W}\right\|^2 + \left\|\hat{W}\right\|^2 - \left\|W^*\right\|^2 \geq \left\|\tilde{W}\right\|^2 - \left\|W^*\right\|^2 \\
|\pi_l s| \leq & \frac{s^2}{2\varepsilon g} + \frac{\varepsilon}{2}\pi_l^2 g \leq \frac{s^2}{2\varepsilon g} + \frac{\varepsilon}{2}\pi_l^2 \overline{g} \\
|\pi_l| \leq & |\pi_0|
\end{cases}
$$

(23)

Therefore, Equation (24) can be acquired by Equations (22) and (23):

$$\dot{V} \leq -\frac{s^2}{2\varepsilon g} - \frac{\sigma}{2}\left\|\tilde{W}\right\|^2 + \frac{\varepsilon}{2}\pi_0^2 \overline{g} + \frac{\sigma}{2}\left\|W^*\right\|^2.$$

(24)

Equation (25) can be realized by Equation (24) because of $\tilde{W}^T \Gamma^{-1} \tilde{W} \leq \overline{\gamma}\left\|\tilde{W}\right\|^2$ ($\overline{\gamma}$ is the maximum eigenvalue of $\Gamma^{-1}$).

$$\dot{V} \leq -\frac{1}{\alpha_0}V + \frac{\varepsilon}{2}\pi_0^2 \overline{g} + \frac{\sigma}{2}\left\|W^*\right\|^2$$

(25)

where $\alpha_0 = \max\{\varepsilon, \overline{\gamma}/\sigma\}$.

According to Lemma B.5 in [33], the inequality in Equation (25) is solved as

$$
\begin{aligned}
V &\leq e^{-t/\alpha_0}V(0) + \left(\frac{\varepsilon}{2}\pi_0^2 \overline{g} + \frac{\sigma}{2}\left\|W^*\right\|^2\right)\int_0^t e^{-(t-\tau)/\alpha_0}d\tau \\
&\leq e^{-t/\alpha_0}V(0) + \alpha_0\left(\frac{\varepsilon}{2}\pi_0^2 \overline{g} + \frac{\sigma}{2}\left\|W^*\right\|^2\right), \ \forall t \geq 0
\end{aligned}
$$

(26)

According to Equation (21), $V(0)$ is a specific value at the initial time, similar to constant $C_1$. Thus, Equation (26) can be rewritten as

$$V \leq C_1 e^{-t/\alpha_0} + C_2, \ \forall t \geq 0$$

(27)

where $C_2 = \alpha_0\left(\frac{\varepsilon}{2}\pi_0^2 \overline{g} + \frac{\sigma}{2}\left\|W^*\right\|^2\right)$.

The first term in Equation (27) gradually decays to $0^+$ over time. For this reason, as long as the second term is guaranteed to be a very small amount, it can be guaranteed that when $t \to \infty$, $V \to 0^+$.

Therefore, the control system remains stable as long as $\alpha_0 \to 0^+$.

## 4. Computer Simulation Experiment Results and Analysis

The USV named "Lanxin" [29] was adopted for the computer simulation. With a speed of 8.5 knots, the parameters of the Nomoto model are listed as $K = 1.502$, $T = 0.905$, $\alpha_1 = 0.001$, and $\alpha_{i(i=1,2,3\cdots)} = 0$. Considering the disturbances of wind and waves, the equivalent interference model can be replaced by a transfer function of a second-order wave driven with white noise [34]:

$$\delta_D = z(s) \cdot w(s) \tag{28}$$

where $w(s)$ is Gaussian white noise, whose average value is zero, and the power of spectral density is 0.1; $z(s)$ is a transfer function of the second-order wave and is defined by Equation (29):

$$\begin{cases} z(s) = \frac{k_\omega \cdot s}{s^2 + 2\xi\omega_0 s + \omega_0^2} \\ k_\omega = 2\xi\omega_0\sigma_m \end{cases} \tag{29}$$

where $\omega_0$ is the wave's dominant frequency, $\xi$ is the damping coefficient, and $k_\omega$ and $\sigma_m$ are constants.

### 4.1. Verification of Practical Performance Experiments

The practical performance of the intelligent control algorithm is confirmed though five computer simulation experiments. In the five experiments, the control parameter of the sliding mode surface is $\lambda = 1.45$, $\eta = 0.5$, the parameters of the IGA are $N = 30$, $p_{mh} = 0.5$, $p_{ml} = 0.04$, $k_1 = 2.4$, $k_2 = 0.9$, and the parameter of the crossover operation is $p_c = 0.85$. The excitation function of the hidden layer is $f_1(x) = \frac{2}{1+e^{-x}} - 1$, the excitation function of the output layer is $f_1(x) = 10 \cdot \left( \frac{2}{1+e^{-x}} - 1 \right)$, and the learning parameters of the RBF network are $\alpha = 0.05$ and $\eta = 0.85$.

During the experiment, the optimization process tended to converge when the optimization process executions exceeded seven. At the same time, in order to facilitate data calculation, an odd number of experiments was performed. Thus, the first nine sets of experiments were intercepted here, as shown in Figure 4. Compared with the general GA, the IGA optimization process converged faster, and the objective function value was smaller. As shown in Table 1, the average value of the objective function of GA optimization is 974.33, and the average/median value is 0.998. The average value of the objective function of IGA optimization is 821.44, which is 15.7% smaller than that for the GA; the average/median value is 0.975, which is 2.3% smaller than that of the GA.

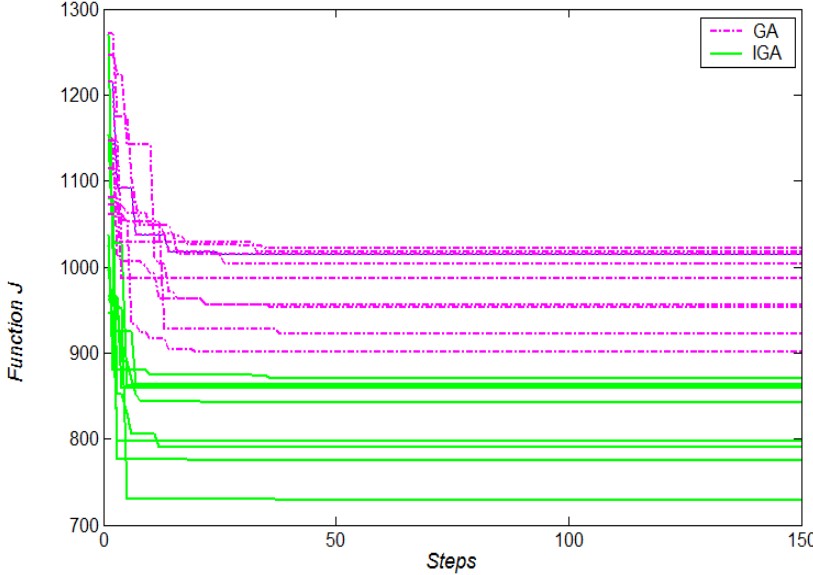

**Figure 4.** The optimization process of the cost function of the improved genetic algorithm (IGA).

**Table 1.** The minimum of objective function.

| Algorithms | 1 | 2 | 3 | 4 | 5 | 6 | 7 | 8 | 9 | Average | Average/Median |
|---|---|---|---|---|---|---|---|---|---|---|---|
| GA | 902 | 922 | 958 | 960 | 976 | 1003 | 1011 | 1022 | 1015 | 974.33 | 0.998 |
| IGA | 740 | 775 | 790 | 800 | 842 | 854 | 858 | 861 | 873 | 821.44 | 0.975 |

The specific results of the ship motion control under various experimental conditions are shown in Table 2. Table 2, A represents the heading change experiment. In this experiment, the controller parameters were set as described above. The first order response pattern was used to perform the tracking experiments. The amplitude was 030°, and the initial value of the experiment was 000° without interference. The test results are shown in Table 2, row A.

B represents the model perturbation experiment. The difference here from experiment A is the added perturbation of the system parameters and that the parameter values of USV are perturbed by 40%. The test results are shown in Table 2, row b.

C represents the sinusoidal interference experiment. The difference here from experiment A is the added external sinusoidal interference. The system output is subjected to sinusoidal interference with an amplitude of 2° and a frequency of 0.1 rad/s. The test results are shown in Table 2, row C.

D represents the white noise interference experiment. The difference here from experiment C is that the external interference changes to white noise. The system output is subjected to white noise with an amplitude of 0.1. The test results are shown in Table 2, row D.

E represents the compound interference experiment. The difference here from experiment A is the added compound external interference. The parameter values of USV are perturbed by 40%, and the system output is subjected to white noise with an amplitude of 0.1. The test results are shown in Table 2, row E.

Based on the result of the experiments in Table 2, the effectiveness and practicality of the intelligent control algorithm is verified because the control performance indicators satisfied the engineering design requirements.

**Table 2.** The control performance indicators of the intelligent control algorithm of the USV.

| Experiments | Stabilization Time | Overshoot | Chattering |
|---|---|---|---|
| A | 65 s | 0 | 0 |
| B | 68 s | 0 | 0 |
| C | 70 s | 0.8% | 0 |
| D | 75 s | 1.0% | 1% |
| E | 80 s | 1.2% | 2% |

*4.2. Verification of Advanced Performance Experiments*

In order to verify the advanced performance of the intelligent control algorithm of USV based on the IGA-optimized RBF neural network proposed in this paper, the tracking control results of this method are compared with the results of the sliding mode control algorithm based on the RBF neural network and the algorithm based on a fuzzy neural network. The three parameters of control performance are then compared between the three methods.

In this paper, three algorithms are applied to the navigation control of the USV under certain conditions in the following three ways. First, the initial course is 000°, and the tracking course is 030°. Second, the second-order pop model is used to determine the compound interference of wind and waves. Third, a servo-driven model of a steering gear is included in the USV control model.

The navigation control of the USV is then determined using the three algorithms mentioned above, and the results of the comparison experiment are shown as Figure 5.

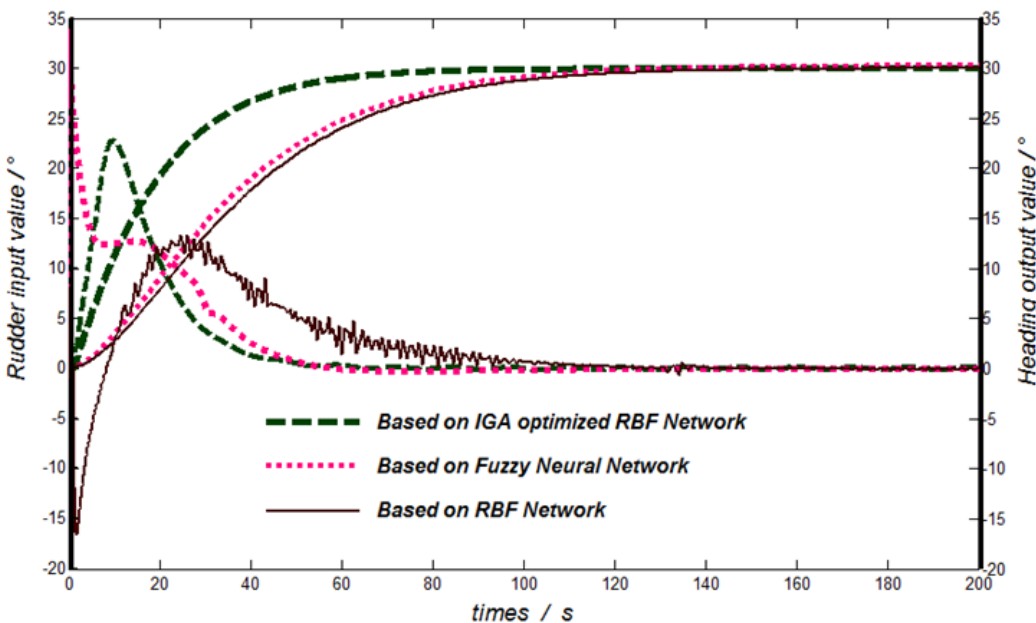

**Figure 5.** Results of the comparison experiment.

## 5. Analysis and Discussion of Results

### 5.1. Control Performance of the Intelligent Algorithm Based on an IGA-Optimized RBF Network

As can be seen from Table 2, the maximum value of the heading stabilization time is initially 80 s, and its tracking speed is fast. After 80 s, the heading tracking output value remains unchanged, so the tracking performance of the control system is stable and reliable. Second, the maximum value of the system's overshoot is 1.2% (less than 5%), which meets actual engineering standards. Third, the maximum value of the control system input's chattering is 2%, which is ideal.

This shows that the intelligent control algorithm of USV based on the IGA-optimized RBF neural network proposed in this paper can effectively track the target course, and its tracking accuracy is high.

### 5.2. Comparison of USV Navigation Control

Figure 5 shows that, under the same compound interference situation, the stability of the intelligent control algorithm based on the IGA-optimized RBF neural network is the fastest, and its control chatter is also the weakest. In practice, the weaker the chattering, the more it can reduce the load on the steering gear and thus protect it.

In Table 3, it shows that the control performance of the intelligent control algorithm based on the IGA-optimized RBF neural network is better than the control performance of the sliding mode control algorithm based on the fuzzy neural network. Moreover, the control performance of the sliding mode control algorithm based on the fuzzy neural network is better than the control performance of the sliding mode control algorithm based on the RBF neural network. Compared with the other two methods, the intelligent control algorithm for USV based on the IGA-optimized RBF neural network can be applied to the research and development of UAS control systems.

**Table 3.** The typical control performance indicators of the three algorithms.

| Control Algorithms | Stabilization Time | Overshoot | Chattering |
| --- | --- | --- | --- |
| Based on RBF Network optimized by IGA | 80 s | 1.2% | Weak |
| Based on the Fuzzy Neural Network | 120 s | 1.5% | Little |
| Based on the RBF Network | 130 s | 2.0% | Medium |

## 6. Conclusions

When an RBF neural network is used for the modeling and control of a motion system, it is easy to fall into the local extreme since the parameters mainly depend on experience; thus, it is still difficult to apply the RBF to such a design. In response, this article applied the IGA to optimize the RBF network for effective application in the development of UAS control systems.

Through computer simulation experiments, we found that: (1) The intelligent control algorithm of the USV based on the IGA-optimized RBF neural network proposed in this paper can effectively track the target course, and its tracking accuracy is high; (2) compared with the sliding mode control algorithm based on a Fuzzy neural network and RBF network, the intelligent control algorithm based on the IGA-optimized RBF neural network is more advanced and can thus be applied to the research and development of UAS control systems.

In addition, it should be noted that the improvement of the GA in this paper is performed by modifying its adaptation value and mutation probability. There are possibly better ways to improve the GA. At the same time, there are no observational data on external interference, which, to a certain extent, can cause the chattering problem for the control input.

**Author Contributions:** R.W. was responsible for writing and revising the topics of this article, including the preliminary research design, data collection, and data analysis. D.L. was responsible for managing and coordinating responsibilities and implementing research activity plans, including literature retrieval and data collection. K.M. supervised and led the planning and execution of the research activities, including the preliminary research design, data collection, and icon design. All authors have read and agreed to the published version of the manuscript.

**Funding:** This work was supported by the Natural Science Research Project of Universities in Jiangsu Province under Grant No. 19KJD580001, 19KJA150005, and 18KJB580003, and the Key R&D Program Projects in Hainan Province under Grant No. ZDYF2018019.

**Conflicts of Interest:** There is no conflict between authors, and the research content is shared between them. The funding agencies were the Hainan Provincial Department of Science and Technology and Jiangsu Provincial Department of Education, but neither participated in designing the content of the article. The authors agreed to submit the manuscript to the Journal of Marine Science and Engineering. The funding agencies did not affect the submission of the manuscript. There are no other conflicts.

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
