# Peer review of "Optimized Radial Basis Function Neural Network Based Intelligent Control Algorithm of Unmanned Surface Vehicles"

_jmse, doi:10.3390/jmse8030210_

Round 1

Reviewer 1 Report

The paper proposes a GA+RBFNN+SMC mechanism for USV control.

The work could be of interest but, in my opinion, several aspects require additional clarification or improvement.

My main concern about this paper is related to contribution. After reading the document, it's not clear to me which parts of the work have the authors developed for this work and which ones have been taken from their previous publications or from other researchers'.

Another important aspect I consider that should be improved is the experimental validation as a support for the conclusions.

In general, the document is hard to read and additional English writing revision is required.

More in detail, my reviewing suggestions are the following ones:

  • A specific paragraph explaining clearly the contribution of the paper should be added, for example in the introduction section.
  • There is a lack of reference to previous GA+RBFNN proposals and, specially, to GA+RBFNN+SMC algorithms and architectures (https://www.google.com/search?q=diagram+of+ga-asmc-rbf+control&client=firefox-b-d&tbm=isch&source=iu&ictx=1&fir=4YfkN7Htv_IKrM%253A%252CRsI3wG_iYJd_dM%252C_&vet=1&usg=AI4_-kR3nd_sPiC6C6seE-nFzURLiqyF-A&sa=X&ved=2ahUKEwipoOzNofHnAhULohQKHTniAl0Q9QEwAnoECAkQCQ#imgrc=4YfkN7Htv_IKrM)
  • In line 63 "Since the mapping from input to output is non-linear, the mapping from hidden layer space to output space is linear" can be read as a cause-consequence, which is not certain.
  • In general, the explanation of the equations should be put after and not before them, or at least change the text to differenciate when a symbol has already appeared or is going to appear later (e. g. lines 73 or 76).
  • A RBFNN+GA bibliography reference should be added in line 82.
  • Fig. 2 include references to fit values that are described in the text with alternative names. The standard terminology should be used.
  • Reference missing in line 93?
  • "N" is used for population size, generations, approximation step.
  • I understand that standard elitism is the procedure described in line 108.
  • Similar schemes to fig.3 should be referenced, and the main differences explained.
  • Line 127 gyration and followability index nomenclature is not commonly used.
  • Reference missing in line 159.
  • Error in line 166.
  • Reference missing in line 174, unless this is proposed by the authors.
  • Lines 129 and 177, D not described.
  • Reference missing in line 181, unless this is proposed by the authors.
  • Line 195, no V'<0 proved.
  • The 5 experiments should be better introduced and their utility justified.
  • Line 208 says "For one of the experiments...", but the parametrization of all of them should be commented.
  • Figure 4 looks to be a particular case, but average/median behaviour should be presented.
  • Figure 5 shows the result of a particular experiment. More general statistically significant results should be provided.
  • Also in figure 5, the proposed control shows a first order response pattern while the alternative methods look like a second order system.
  • Differences between this works and reference [18] should be indicated in the paper.

Author Response

Response to Reviewer 1 Comments

-----------------------------------------------------------------------------------------Point 1: A specific paragraph explaining clearly the contribution of the paper should be added, for example in the introduction section.

Response 1: The paragraph about contribution has been added, and marked in Yellow.

Point 2: There is a lack of reference to previous GA+RBFNN proposals and, specially, to GA+RBFNN+SMC algorithms and architectures.

Response 2: We searched the bibliography and references and found some proposals about GA + RBFNN.

Point 3: In line 63 "Since the mapping from input to output is non-linear, the mapping from hidden layer space to output space is linear" can be read as a cause-consequence, which is not certain.

Response 3: This sentence is not a causal structure, but a side-by-side structure. The result is the sentence after “so that”. It has been marked in Yellow.

Point 4: In general, the explanation of the equations should be put after and not before them, or at least change the text to differenciate when a symbol has already appeared or is going to appear later (e. g. lines 73 or 76).

Response 4: The explanation of the equation (2) has been changed, marked in Yellow.

Point 5: A RBFNN+GA bibliography reference should be added in line 82.

Response 5: Some references have been added in, and the content of this paragraph has also been supplemented. Please refer to the sentence marked in Yellow.

Point 6: Fig. 2 include references to fit values that are described in the text with alternative names. The standard terminology should be used.

Response 6: It has been revised.

Point 7: Reference missing in line 93?

Response 7: Reference [15] has been added.

Point 8: "N" is used for population size, generations, approximation step.

Response 8: Duplicate definition, sorry. It has been revised “G” and marked in Yellow.

Point 9: I understand that standard elitism is the procedure described in line 108.

Response 9: The description was not clear, it was revised and marked in Yellow.

Point 10: Similar schemes to fig.3 should be referenced, and the main differences explained.

Response 10: The content of Figure 3 has been explained by a new paragraph marked in Yellow.

Point 11: Line 127 gyration and followability index nomenclature is not commonly used.

Response 11: It has been revised as “the ship's parameters”.

Point 12: Reference missing in line 159.

Response 12: Reference [16] has been cited.

Point 13: Error in line 166.

Response 13: The error “W^” has been corrected to “W*”.

Point 14: Reference missing in line 174, unless this is proposed by the authors.

Response 14: The “xita*sgn(s) ” added is going to suppress the external disturbance of the system, where, xita>=dmax.

Point 15: Lines 129 and 177, D not described.

Response 15: D has been revised as dmax.

Point 16: Reference missing in line 181, unless this is proposed by the authors. Response 16: Lyapunov Function proposed in paper was going to guarantee the stability of the control system.

Point 17: Line 195, no V'<0 proved.

Response 17: In order to guarantee the stability of the control system, Some paragraphs are added and marked in Yellow. Please see it in attachment.

Point 18: The 5 experiments should be better introduced and their utility justified.

Response 18: The 5 experiments have been introduced, and added.

Point 19: Line 208 says "For one of the experiments...", but the parametrization of all of them should be commented.

Response 19: This part has been modified.

Point 20: Figure 4 looks to be a particular case, but average/median behaviour should be presented.

Response 20: From your point of view, we have made additions marked in Yellow, in the text.

Point 21: Figure 5 shows the result of a particular experiment. More general statistically significant results should be provided.

Response 21: Nowadays, for general commercial ships, the maximum value of the automatic control device of the ship is 10 ° each time. If the demand exceeds 10 °, it is carried out in stages, 10 ° / time. So, we set a change of 30 °, just for verification. For the research and development of USV, it may not be representative yet, which needs further discussion with ship manufacturers.

Point 22: Also in figure 5, the proposed control shows a first order response pattern while the alternative methods look like a second order system.

Response 22: Indeed a first order response pattern.

Point 23: Differences between this works and reference [18] should be indicated in the paper.

Response 23: In order to indicate the Differences between this works and “reference [18]”, “reference [18]” has been changed as “reference [12]”, and a passage is added, marked in Yellow.

In addition, by the comments of other experts, some parts were revised.

Please refer to the attachment. Thanks for your reviewing!

Reviewer 2 Report

This paper presents an intelligent control algorithm based on Genetic Algorithm optimized Radial Basis Function for tracking stability control of Unmanned Surface Vehicle. The introduction states the purpose of the paper, and the relation between the paper and the previous works is clearly explained. The USV controller is designed with SMC technology and Lyapunov stability theory. The feasibility of the presented algorithm is illustrated via numerical simulations.

The following are my specific comments:

(1) It is stated in Introduction that “the connection weight and threshold parameters are mainly determined by experience”. In Section 2, it is specified that “the IGA is used to determine the connection weight parameters of RBF network”. I wonder how the threshold parameters are determined in the presented algorithm. In addition, it is better to provide a brief explanation for Figure 2.

(2) In Section 2, I cannot find pnh in equation (4). Why is there a “100” in equation (5)? How is the approximation error e(i) calculated? Try to give some explanations.

(3) In Section 3, I cannot find the definitions for the denotations D, H, d(t) and yd. There may be some typo errors, such as “The above control rate” and “it can be pushed”. Please check the explanations below equation (8).

(4) As the system function and the gain are unknown, the optimized RBF neural network is used to approximate the control law. For clarity, it is better to provide a brief description about the learning process of the neural network for the considered system.

(5) The condition to guarantee the stability of the control system should be clearly specified.

(6) In Section 4, I cannot find the definitions for the denotations alfa and pc. Try to provide some remarks about the effects of the control parameter and the learning parameters.

(7) The proposed algorithm is compared with the sliding mode control algorithm based on RBF neural network and the algorithm based on fuzz neural network. To illustrate the efficiency of the proposed algorithm, the authors are suggested to make more comparison with other existing algorithms in literature.

(8) I suggest that both advantages and weak points of the proposed algorithm are specified in Conclusions.

Author Response

Response to Reviewer 2 Comments

-----------------------------------------------------------------------------------------Point 1: It is stated in Introduction that “the connection weight and threshold parameters are mainly determined by experience”. In Section 2, it is specified that “the IGA is used to determine the connection weight parameters of RBF network”. I wonder how the threshold parameters are determined in the presented algorithm. In addition, it is better to provide a brief explanation for Figure 2.

Response 1: It should be the three parameters (connection weights, center width and center value of the Gaussian function). It has been revised and marked in green. In addition, a brief explanation for Figure 2 is provided. Please see it in attachment.

Point 2: In Section 2, I cannot find pnh in equation (4). Why is there a “100” in equation (5)? How is the approximation error e(i) calculated? Try to give some explanations.

Response 2: Not pnh, but pmh in equation (4), it has been revised and marked in green. In equation (5), the selected “100” is a subjective choice. Considering that the value of the approximation error is usually small, the magnification of 100 times is convenient for error identification and is conducive to genetic operations. The absolute error e(i) calculated by xd-x1 in my Simulation program.

Point 3: In Section 3, I cannot find the definitions for the denotations D, H, d(t) and yd. There may be some typo errors, such as “The above control rate” and “it can be pushed”. Please check the explanations below equation (8).

Response 3: Denotation D means the Maximum of disturbance (d(t)), and it has been changed to dmax. H means the non-linear function of . yd means command order of output (y=x1), and it has been revised as xd. “The above control rate” and “it can be pushed” are also revised. These are marked in green. Please see it in attachment.

Point 4: As the system function and the gain are unknown, the optimized RBF neural network is used to approximate the control law. For clarity, it is better to provide a brief description about the learning process of the neural network for the considered system.

Response 4: A brief description is marked in green about the learning process of the neural network for the considered system. Please see it in attachment.

Point 5: The condition to guarantee the stability of the control system should be clearly specified.

Response 5: Some paragraphs are added and marked in green. Please see it in attachment.

Point 6: In Section 4, I cannot find the definitions for the denotations alfa and pc. Try to provide some remarks about the effects of the control parameter and the learning parameters.

Response 6: The remarks about alfa and pc are marked in green. Please see it in attachment.

Point 7: The proposed algorithm is compared with the sliding mode control algorithm based on RBF neural network and the algorithm based on fuzzy neural network. To illustrate the efficiency of the proposed algorithm, the authors are suggested to make more comparison with other existing algorithms in literature.

Response 7: The reason why this paper is compared with the above two methods is that the RBF neural network or its improved network by fuzzy model is integrated into the sliding mode control. From the perspective of control algorithm research, we fully accept your suggestions and do detailed comparative research in the follow-up. Thank you!

Point 8: I suggest that both advantages and weak points of the proposed algorithm are specified in Conclusions.

Response 8: We proposed a weak point in Conclusions, and marked in green. Please see it in attachment.

In addition, by the comments of other experts, some parts were revised.

Please refer to the attachment. Thanks for your reviewing!

Reviewer 3 Report

Title: Genetic Algorithm optimized RBF Neural Network based intelligent control algorithm of Unmanned Surface Vehicle
---------------------------------------------------------------------------------------------------------------------
Title needs to be revised. More clear title for the readers

Abstract needs revision. Make it again clearer to readers about contribution of current study to the state of art.

Section I - Paragraph 1 needs more references on recent developements in USV like:

- Singh, Y., S. Sharma, R. Sutton, and D. Hatton. "Optimal path planning of an unmanned surface vehicle in a real-time marine environment using a dijkstra algorithm."
In Marine Navigation, pp. 399-402. CRC Press, 2017.

- Bibuli, Marco, Yogang Singh, Sanjay Sharma, Robert Sutton, Daniel Hatton, and Asiya Khan. "A two layered optimal approach towards cooperative motion planning of unmanned surface vehicles in a constrained maritime environment."
IFAC-PapersOnLine 51, no. 29 (2018): 378-383.

Section II- Reference to RBF required in Para 1

-Why IGA is chosen ? Provide benchmark study.

-Where is the motion model of the USV? It is not clear if a kinetic or dynamic model is used ? If yes, provide proofs?

-Any particular USV is used. If yes, then provide details.

-How is external disturbance described and defined in the manuscript ? Provide a certain paragraph.

- Provide a study in terms of sea surface currents effect on USV motion model.

Author Response

Response to Reviewer 3 Comments

-------------------------------------------------

Point 1:Title needs to be revised. More clear title for the readers

Response 1: It has been revised and marked in red. Please see it in attachment.

Point 2:Abstract needs revision. Make it again clearer to readers about contribution of current study to the state of art.

Response 2: It has been revised and marked in red. Please see it in attachment.

Point 3:Section I - Paragraph 1 needs more references on recent developements in USV like: - Singh, Y., S. Sharma, R. Sutton, and D. Hatton. "Optimal path planning of an unmanned surface vehicle in a real-time marine environment using a dijkstra algorithm." In Marine Navigation, pp. 399-402. CRC Press, 2017. - Bibuli, Marco, Yogang Singh, Sanjay Sharma, Robert Sutton, Daniel Hatton, and Asiya Khan. "A two layered optimal approach towards cooperative motion planning of unmanned surface vehicles in a constrained maritime environment." IFAC-PapersOnLine 51, no. 29 (2018): 378-383.

Response 3: It has been revised and marked in red along with other citations. Please refer to the attachment.

Point 4:Section II- Reference to RBF required in Para 1

Response 4: It has been revised and marked in red along with other citations. Please refer to the attachment.

Point 5:-Why IGA is chosen ? Provide benchmark study.

Response 5: It has been revised and marked in red along with other explanations. Please refer to the attachment.

Point 6:--Where is the motion model of the USV? It is not clear if a kinetic or dynamic model is used ? If yes, provide proofs?

Response 6: It has been revised by Nomoto’s equation cited as a dynamic mathematical model, and marked in red. Please refer to the attachment.

Point 7:-Any particular USV is used. If yes, then provide details.

Response 7: In Section 4, USV named “Lanxin” is just cited, it’s parameters are adopted in paper, and marked in red.

Point 8:-How is external disturbance described and defined in the manuscript ? Provide a certain paragraph.

Response 8: In Section 4, disturbance of wind and wave is cited with the equivalent interference model, and marked in red.

Point 9:- Provide a study in terms of sea surface currents effect on USV motion model.

Response 9: In view of sea surface currents only affect the position of the USV, it is not studied in paper.

In addition, by the comments of other experts, some parts were revised.

Thanks for your reviewing!

Round 2

Reviewer 1 Report

The authors have attended most of the reviewing comments.

In my opinion, and in order to be published, some minor aspects still could be corrected in the paper:

  • After inserting new references in line 44, the paragraph is now a bit confusing, and line 49 is not properly written.
  • Lines 64 and 67 also include some typos or misplaced capitals.
  • Line 112 looks incorrectly written.
  • Figure 2 could be rearranged to avoid line crossing.
  • Line 170 should be rewritten, since it's confusing.
  • Line 271 should be reordered.
  • Line 278, why just 9 comparisons?
  • Line 288 all experiments repeat the same sentence about 1st order pattern.
  • Line 344 > symbols should be replaced by text.
  • Line 361 "only by" expression should be replaced to highlight paper relevance

Author Response

Response to Reviewer 1 Comments

-------------------------------------------------------------------------------------------------

Point 1: After inserting new references in line 44, the paragraph is now a bit confusing, and line 49 is not properly written.

Response 1: In line 44-46, and 49. It has been revised and marked in Yellow.

Point 2: Lines 64 and 67 also include some typos or misplaced capitals.

Response 2: In line 64-65, and 66-67. It has been revised and marked in Yellow.

Point 3: Line 112 looks incorrectly written.

Response 3: In line 110-112. It has been revised and marked in Green.

Point 4: Figure 2 could be rearranged to avoid line crossing.

Response 4: Please refer to Fig.2. . It has been rearranged.

Point 5: Line 170 should be rewritten, since it's confusing.

Response 5: In line 170-171. It has been revised and marked in Yellow.

Point 6: Line 271 should be reordered.

Response 6: In line 270-271. It has been revised and marked in Yellow.

Point 7: Line 278, why just 9 comparisons?

Response 7: In line 277-279. A new paragraph has been added for explanation. Marked in yellow.

Point 8: Line 288 all experiments repeat the same sentence about 1st order pattern.

Response 8: In line 293-294, 296-297, 299-300, and 302-303. It has been modified, and marked in Yellow.

Point 9: Line 344 > symbols should be replaced by text.

Response 9: In line 340-344. It has been revised and marked in Yellow.

Point 10: Line 361 "only by" expression should be replaced to highlight paper relevance.

Response 10: In line 359-360. It has been revised and marked in Yellow.

In Round 1. Your comments have inspired me a lot, especially in the design framework.

Thank you! Thanks for your second reviewing!

Reviewer 2 Report

The quality of the manuscript is improved after the revision. Nevertheless, some minor issues need to be addressed before the paper can be published. For example, “linear or nonlinear Yes, discrete or continuous” in Section 2.2.

Author Response

Point 1: The quality of the manuscript is improved after the revision. Nevertheless, some minor issues need to be addressed before the paper can be published. For example, “linear or nonlinear Yes, discrete or continuous” in Section 2.2.

Response 1: In response to this problem, we adjusted it as follows: For any form of objective function and constraint optimization problem to be solved, whether it is linear or non-linear, discrete or continuous, the genetic algorithm does not require any mathematical model. Due to its evolutionary nature, the inherent nature of the problem is not needed during the search process. Marked in Yellow.

Thanks again for your review!

Reviewer 3 Report

The authors have made the changes required by the reviewers and publication looks good to be published.

Author Response

Thank you!

Thanks again for your review!